# Association between a soy-based infant diet and the onset of puberty: A systematic review and meta-analysis

**Flávia Ramos Kazan Oliveira**◉[☉], **Ana Flora Silva e Gustavo**[☉], **Renan Braga Gonçalves**[☉], **Fernanda Bolfi, Adriana Lúcia Mendes, Vania dos Santos Nunes-Nogueira**◉*

Department of Internal Medicine, São Paulo State University (UNESP), Medical School, Botucatu, Sao Paulo, Brazil

☉ These authors contributed equally to this work.
* vania.nunes-nogueira@unesp.br

**Data Availability Statement:** All relevant data are within the paper and its Supporting Information files.

## Abstract

The objective of this systematic review was to evaluate the association between a soy-based infant diet and the onset of puberty. We included studies in which children were fed a soy-based diet, and we compared them with those who were not. The primary outcomes were the onset of puberty in girls (thelarche, pubarche, and menarche age), boys (pubarche, voice change, testicular and penis enlargement age), and both (risk of delayed and precocious puberty [PP]). Search strategies were performed in PubMed, Embase, LILACS, and CENTRAL databases. Two reviewers selected eligible studies, assessed the risk of bias, and extracted data from the included studies. The odds ratio (OR) and mean difference (MD) were calculated with a 95% confidence interval (CI) as a measure of the association between soy consumption and outcomes. We used a random-effects model to pool results across studies and the Grading of Recommendations Assessment, Development, and Evaluation to evaluate the certainty of evidence. We included eight studies in which 598 children consumed a soy-based diet but 2957 did not. The primary outcomes that could be plotted in the meta-analysis were the risk of PP and age at menarche. There was no statistical difference between groups for PP (OR: 0.51, 95% CI: 0.09 to 2.94, 3 studies, 206 participants, low certainty of evidence). No between-group difference was observed in menarche age (MD 0.14 years, 95% CI -0.16 to 0.45, 3 studies, 605 children, low certainty of evidence). One study presented this outcome in terms of median and interquartile range, and although the onset of menarche was marginally increased in girls who received a soy-based diet, the reported age was within the normal age range for menarche. We did not find any association between a soy-based infant diet and the onset of puberty in boys or girls.

**Trial Registration**: PROSPERO registration: CRD42018088902.

## Introduction

As a consequence of the interaction of genetic, endocrine, and environmental factors in the last four decades, the onset of puberty in girls has occurred earlier [1]. Certain components in

**Funding:** The author(s) received no specific funding for this work.

**Competing interests:** The authors have declared that no competing interests exist.

the diet and environment are affecting the endocrine system. This has consequently led to the early development of secondary sexual characteristics in both girls and boys.

Efforts to implement healthy eating habits that are associated with the possible benefits of soy when consumed early in life have resulted in an increase in the consumption of soy products [2,3]. Additionally, soy foods have been the base of supplements, infant formulas, flour, milk, juices, soy sauce, tofu, and they are included in many industrialized foods that are consumed predominantly in childhood. Moreover, these foods are important sources of polyphenols, also called isoflavones, that can function as estrogen agonists/antagonists or selective estrogen receptor (ER) modulators.

These ERs can interact with a wide variety of compounds. Despite their low binding affinities, isoflavones exhibit estrogenic activities. Therefore, these compounds are referred to as phytoestrogens and can interfere with the metabolism of steroids [4]. This binding effect has a higher affinity for ER beta than for ER alpha. Sometimes, they are classified as natural selective estrogen receptor modulators (SERMs)—mixed estrogen agonists/antagonists [5]. Isoflavones, and in particular genistein, have the potential to exert physiological effects. They affect the signal transduction pathways and inhibit the activity of many enzymes (e.g., protein kinase tyrosine, kinase activated by mitogens, and DNA topoisomerase). They also regulate the cellular factors that control cell growth and differentiation [6,7]. Isoflavones are often classified as endocrine disruptors, that is, chemicals capable of altering the function of the endocrine system and potentially resulting in adverse health effects [8]. Because of their hormonal activity, it is concerning that these phytoestrogens might promote the onset of early puberty.

Experimental animal studies have shown that xenobiotics, such as polychlorinated biphenyls (PCBs), phytoestrogens, fungicides, and pesticides may affect sexual differentiation [9]. This interference has a high probability of affecting both reproductive physiology and behavior at several stages of life [10]. In female rats, early exposure (late embryonic and/or early postnatal) to low doses of PCBs [10] or soy significantly affected mating behavior [11].

Genistein, the main phytoestrogen in soy, has a wide range of biological activities. It binds to ER alpha and ER beta, although it also has an antiestrogenic action [12]. At low concentrations, genistein acts as an estrogen and has an inhibitory effect on lipogenesis. There are also sexual differences in the effect of genistein on adipose deposition and insulin resistance, an effect that involves ER beta [13]. At higher concentrations, genistein promotes lipogenesis through the PPAR gamma pathway and the ER-independent pathway [12]. In female mice, post-weaning dietary genistein consumption advanced puberty by decreasing the age of the vaginal opening, increased the length of the estrus stage, and accelerated mammary gland development [14].

A few studies have evaluated the effects of isoflavones on sexual development in humans. The literature reports controversial results regarding their influence on the onset of puberty. According to Cheng et al., girls with a high intake of isoflavones (423.4–19,178 μg/day) in the pre-pubertal period had breast development of Tanner stage 2. This was 0.7 years later than girls whose diet was in the lowest isoflavone tertile intake [15]. Conversely, when exclusively observing male adolescents, Segovia-Siapco et al. concluded that moderate/high consumption (3–20 mg and >20 mg/day) of isoflavones lead to an earlier puberty [16]. In another study, only with adolescent girls, the same authors found no association between isoflavone intake and the age of menarche onset [1]. Strom et al. found no adverse effects on the reproductive health of adults who consumed soy milk during childhood [17]. The American Academy of Pediatrics does not recommend restricting the consumption of soy-based formulas in childhood. They claim that there is a low affinity of soy phytoestrogens to ERs and a low estrogenic potency in bioassays [18]. Thus, any possible effect of the abundance of phytoestrogens in an infant's diet would be balanced by the low affinity of these compounds to ERs.

Considering this conflicting evidence in the literature, this study aimed to evaluate the association between a soy-based infant diet and the onset of puberty in girls and boys.

## Materials and methods

A systematic review was conducted and reported according to the Preferred Reporting Items for Systematic Reviews and Meta-Analyses (PRISMA) Statement [19]. The protocol was registered in the International Prospective Register of Systematic Reviews (CRD42018088902).

### Eligibility criteria

We included observational (cohort and cross-sectional studies) and randomized controlled trials (RCT) that met the PECO structure described below.

**Participants (P).**   The participants were children, adolescents, or adults of both sexes from all ethnic backgrounds where it was possible to assess the onset of puberty, presence/absence of precocious puberty, or delayed puberty.

**Exposure (E).**   The exposure group was comprised of individuals who consumed a soy-based diet. We considered a soy-based infant diet to be the consumption of soy-based products (e.g., soy-based formula or milk), higher food intake of isoflavones, or the use of soy protein-based supplements. Given that the mean intake of isoflavones varies from country to country and there is no definition of either a mean or high intake of this phytoestrogen in children, we did not use a cut-off point to separate participants with a high intake from those with a low intake. However, the studies were included if the authors used any tool to classify children as high or low consumers. When the mean daily intake of soy and its' derivatives was not available, we included patients who were known to have a higher consumption of soy than the general population, such as vegetarians and Asian populations. The assessment of soy consumption was determined through dietary records, food frequency questionnaires, or the participant's use of soy supplements.

**Comparison (C).**   We considered participants who did not consume a soy-based infant diet as a comparison group.

**Outcomes (O).**   Primary outcomes were related to the onset of puberty in girls and boys. In girls, this included the onset of breast development (thelarche age), first appearance of pubic hear (pubarche age), and first menstrual cycle (menarche age). In boys, this included the age at onset of testicular growth, age at onset of pubarche, age at onset of voice change, and age at penis enlargement. In both girls and boys, the primary outcomes were the risk of precocious puberty (development of secondary sexual characteristics before the age of 8 years in girls and 9 years in boys) and the risk of delayed puberty (absence of testicular enlargement in boys by 14 years or lack of breast development in girls by 13 years). The secondary outcomes were other indicators of puberty such as height at the last follow-up visit (measurement at the beginning of puberty, during puberty, or as adult height), adverse events, and for boys, age at first ejaculation.

### Exclusion criteria

We excluded studies without a comparison group and studies in which phytoestrogens were not derived from soy. As our objective was to compare the onset of puberty in children who consumed soy with children who did not, we excluded case-control studies wherein participants with early or precocious puberty were compared with those who had soy consumption.

## Identification of the studies

**Electronic databases.** General research strategies were applied to the main electronic health databases: Embase (Elsevier, 1980–2020), Medline (by PubMed), LILACS (by Virtual Health Library), Controlled Clinical Trials of Cochrane Collaboration (CENTRAL), Trip database, SCOPUS, and Web of Science. Databases were searched on December 17, 2017 and updated on April 02, 2020. The Medical Subject Headings that were used included: "Soy Foods," "Soy Milk," "Soybeans," "Soybeans Protein," "Soybean oil," "Puberty," "Puberty, precocious," "Sexual Maturation," "Menstruation Disturbances." The search strategies for the primary databases are included in the Supporting Information (S1 File). We also searched for unpublished studies among dissertations and theses (ProQuest Dissertation & Theses Global, WorldCatDissertations, The Digital Library of Theses and Dissertations of the University of São Paulo, Catalog of Theses & Dissertations-CAPES), ClinicalTrials.gov website, and Brazilian Registry of Clinical Trials (ReBec).

EndNote X9 citation management software was used to download the references and remove duplicate entries. The initial screening of abstracts and titles was performed using the free web application Rayyan QCRI [20].

**Study selection.** Three reviewers (FRKO, AFSG, and RBG) independently selected the titles and abstracts identified during the literature search. The studies selected for full-text review were subsequently assessed for adequacy of the proposed "PECO" structure. In case of disagreement, a final consensus was reached between the reviewers and the project coordinator (VSN-N).

## Data extraction and management

For the studies selected for inclusion, the reviewers used a standardized extraction form. This ensured that all information contained in each study (number of patients, average age, study design, inclusion and exclusion criteria, exposure, unexposed, outcomes analyzed, follow-up time, and risk of bias) could be compared.

## Assessment of risk of bias in the included studies

For each selected randomized study, the risk of bias was evaluated according to the criteria described in the Cochrane Collaboration tool (RoB 1). This tool encompasses seven domains: the process of randomization, concealing allocation, blinding of participants and researchers, blinding of outcome assessors, whether the losses were included in the final analysis, selective reporting of outcomes, and others [21].

For each selected observational study, the risk of bias was evaluated according to the criteria described by the Newcastle–Ottawa Scale (NOS) for cohort studies. This scale encompasses three domains: selection (four items), comparability (one item), and outcome (three items) [22].

## Unit of analysis

The unit of analysis was the data published in the included studies.

## Data analyses

Similar outcomes were plotted in the meta-analysis using Stata Statistical Software 16 (*Stata Statistical Software*: *Release 16*. College Station, TX, StataCorp LLC, USA). Continuous data were expressed as means and standard deviations. Differences between means (MD) with 95% confidence intervals (CIs) were used to estimate the exposure effect. For the risk of precocious puberty, we present a meta-analysis that included and excluded both-armed zero-event studies

[23]. In the first scenario, we added 0.5 to each cell of the 2 × 2 table for the trials with zero events in both arms and the odds ratio (OR) was calculated with a 95% CI [23]. For the second method, we used the Peto one-step OR method [24]. We selected a random-effects model for the meta-analysis. The studies were evaluated separately according to their design (observational versus RCT).

When sufficient studies were available, we conducted sensitivity analyses to assess the robustness of our results. These analyses were performed by comparing studies according to the risk of bias, length of exposure, and assessment of pubertal development through self-report or physical examination.

### Assessment of heterogeneity

Inconsistencies among the study results were ascertained by visually inspecting a forest plot and using the Higgins or $I^2$ statistic, in which an $I^2 > 50\%$ indicated a moderate probability of heterogeneity. When sufficient studies were available, the potential causes of heterogeneity among the studies could be evaluated by subgroup analysis. In case of considerable inconsistency ($I^2 > 50\%$) associated with variation in the direction of association and heterogeneity could not be explained, we did not perform a meta-analysis.

### Quality of the evidence

The quality of evidence for estimating the effect of exposure on outcomes that could be plotted in the meta-analysis was generated in accordance with the Grading of Recommendations Assessment, Development, and Evaluation (GRADE) Working Group [25]. GRADE is a structured process for rating the quality of evidence in systematic reviews or in guidelines for health care [25]. Randomized controlled trials begin as high-quality evidence; however, the confidence in the evidence may decrease if the studies have major limitations that may interfere with the estimates of the treatment effect [25]. These limitations include the risk of bias, inconsistency of results, indirectness of evidence, imprecision, and reporting bias [26]. Conversely, observational studies are initiated with a low certainty of evidence. However, the quality of evidence can increase when studies rigorously present one of the following criteria: the magnitude of the treatment effect is very large, there is evidence of a dose–response relationship, or all plausible biases would decrease the magnitude of the treatment effect [27].

When more than 10 studies were included in the meta-analysis, we used a funnel plot to investigate the presence of publication bias [24].

## Results

After removing duplicates, the search strategies yielded 962 references, and we selected 16 studies for the final examination (Fig 1). Eight studies met our eligibility criteria and were therefore included in this review.

A total of eight studies were excluded from the final examination for the following reasons: one was a literature review, one was a case report, two presented only laboratory outcomes, one was an animal study, one did not describe clinical outcomes related to puberty (only behavioral factors), and in two studies the time of follow-up was insufficient to evaluate the health outcomes [28–35].

### Included studies

We included eight studies, seven observational studies (five were longitudinal studies and two were cross-sectional studies) [1,15–17,36–38], and one RCT [39]. The mean follow-up time

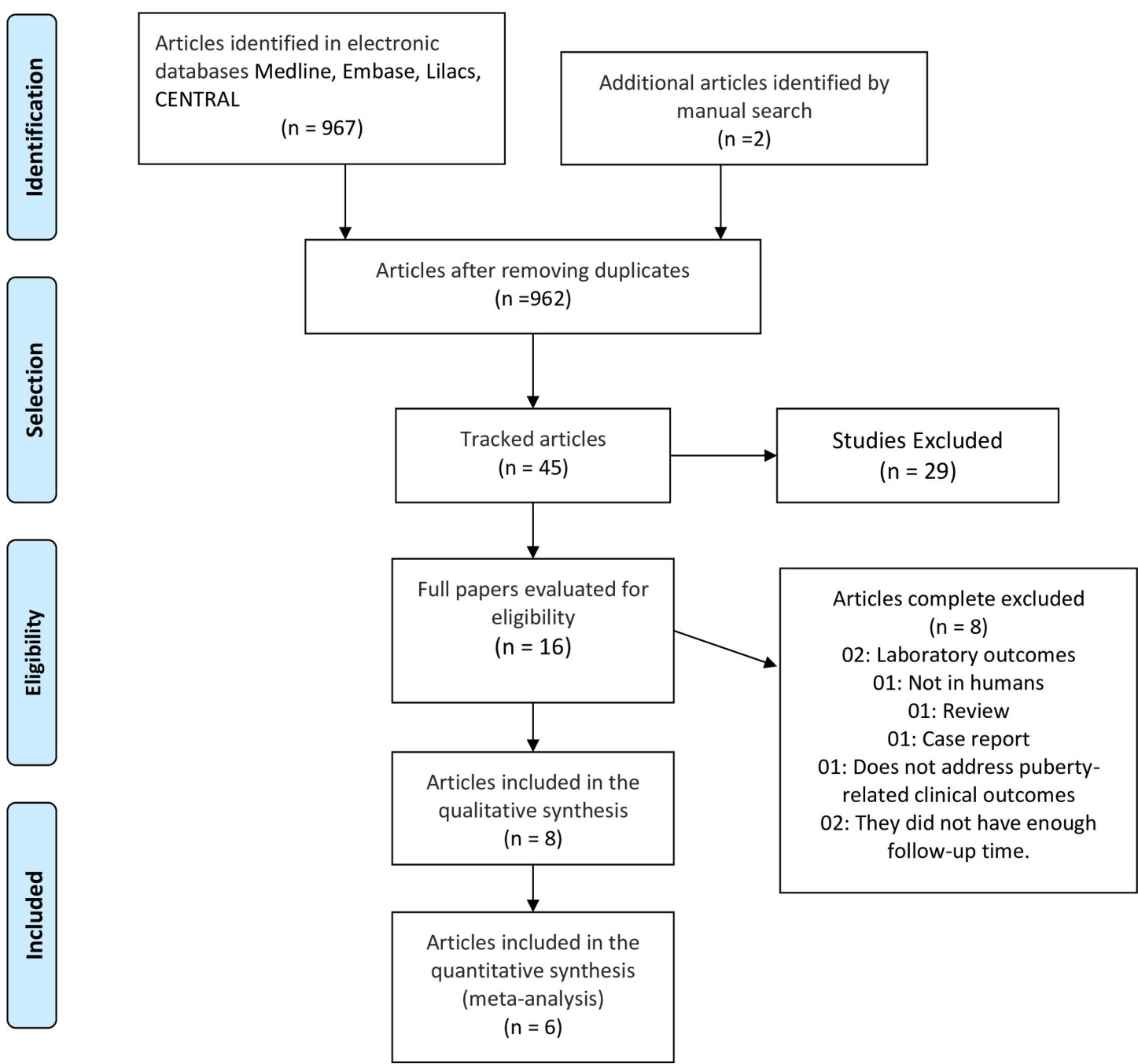

**Fig 1. Flow diagram of the selection of studies.**

was 9.3 years in the longitudinal studies. Regarding the included participants, six studies were conducted with a healthy population, and two studies did not mention if participants with chronic illness were excluded [1,16]. Regarding exposure, four studies used soy-based formula or milk [17,36–38], one used a soy protein supplement [39], and one assessed soy intake by dietary records (the highest and lowest dietary isoflavone tertile were considered exposure and unexposed, respectively) [15]. One study classified soy intake into four categories. We considered unexposed individuals to be those who ate soy foods less than once a week, and as

exposed if they consumed soy products more than three times per day [1]. In the Segovia-Siapco et al. study, we regarded as exposure when the total isoflavone intake was higher than 20 mg/day and as unexposed when the total isoflavone intake was less than 3.0 mg/day. In the Sinai et al. study, the exposure group comprised children with an IgE-mediated cow's milk allergy. They ingested soy-based formula for longer than 3 months, usually at the end of breast-feeding or in combination with breastfeeding. The unexposed group comprised infants considered "healthy" as no gastrointestinal symptoms associated with food were reported.

Regarding potential confounders, two studies did not mention pre-pubertal weight. However, six studies reported that at 7–9 years of age, there was no significant difference between the groups in BMI [15,17,34,36,37,39]. Four studies reported that exposed and unexposed children were similar with respect to birth weight, maternal pre-pregnancy BMI, prenatal smoking status, marital status, and maternal education. However, Cheng et al. reported that unexposed girls were younger at baseline, and they were more likely to be from a smoking household, consume less fruit fiber, and have a lower baseline energy intake. In the Segovia-Siapco et al. studies, there was a significant difference between the groups in terms of site of study (California vs Michigan), type of school (public vs private) and exposure participants were more likely to have parents with a graduate-level education [1,16].

Table 1 shows the main characteristics of the included studies, eligibility criteria, and Table 2 shows outcome results.

**Bias risk assessment.** For the RCT, due to lack of information, the risk of bias was unclear in the seven domains assessed. Table 3 shows the risk of bias in the observational studies. In all the studies, the exposed cohort met our eligibility criteria, and the unexposed cohort originated from the same community as that of the exposed participants. Regarding assessment of exposure, four studies used questionnaires [1,16,17,38], three used food records, and in the RCT, the supplement was provided and supervised by nutritionists [39]. Only one study demonstrated the outcomes of interest at the initiation of the study [15]. All the groups were comparable at baseline. Pubertal development was assessed by a physician in three studies [36,37,39], it was self-reported in four [1,16,17,38], and by nurses in one [15]. In three studies, the follow-up period was insufficient to assess delayed puberty [36,37,39]. Most studies reported loss of follow-up; thus, the data on outcomes for these participants could not be assessed.

**Studies included in the meta-analysis.** *Risk of precocious puberty*. Regarding the risk of precocious puberty, there was no significant difference between the children who were exposed or unexposed (OR: 0.51, 95% CI: 0.09–2.93, 3 studies, 206 participants, low certainty of evidence, Fig 2, Table 4). Excluding the studies with no events in both arms, no difference between groups remained (OR: 0.55, 95% CI: 0.08–3.72, 1 study, 89 participants, low certainty of evidence, supporting information, S1 Fig).

*Age at menarche*. Regarding age at menarche, the meta-analysis of three studies showed no difference between the groups for the onset of menarche (MD 0.14 years, 95% CI -0.16 to 0.45 years, 605 children, low certainty evidence, Fig 3, Table 4). One study presented data of age at menarche in median and interquartile range (IQR). Although this timing was marginally increased in girls who consumed a soy-based diet as an infant, the reported age was within the normal age range for menarche (12.4 years; IQR 11.6 to 13.25; 54 girls) [38].

*Height in the last follow-up*. Four studies evaluated the height of patients at the last visit of the study [17,36,37,39]. Height was assessed at the beginning of pubertal signs in three studies and at an adult age in one study [17]. In the meta-analysis for the Duitama and Strom studies (height measurement at beginning of puberty and at adult age, respectively) no clinically significant difference between groups was found (MD 0.432 cm, 95% CI -1.03 to 1.9 cm, 2 studies, 862 individuals, low certainty evidence, Fig 4, Table 4). The subgroup analysis according to sex

Table 1. Characteristics of included studies.

| Author | Country | Study population | N˚ | Infant diet | | Mean (SD) age at baseline | | Mean (SD) BMI | | Confounders |
|---|---|---|---|---|---|---|---|---|---|---|
| | | | | Exposure | Unexposed | Exposure | Unexposed | Exposure | Unexposed | |
| **Adgent 2012** | UK | White, healthy or with minor problems, girls from the Avon region of the UK, who were expected to deliver between April 1, 1991, and December 31, 1992 | 2,178 | Formula or milk soy-based/ Unknown | Early formula with no soy | Between 8 and 14.5 ys | Between 8 and 14.5 ys | **Age 7–9 years** Ever > 85th Percentile: 26% Never > 85th Percentile: 74% | **Age 7–9 years** Ever > 85th Percentile: 35% Never > 85th Percentile: 65% | Exposure girls were similar to unexposed with respect to birthweight, childhood BMI, as well as maternal pre-pregnancy BMI, prenatal smoking status, marital status, and prenatal vegetarian diet. |
| **Cheng 2010** | Germany | Healthy, white, children | 79 | Isoflavone intake[a] Girls: 1,199 μd/d Boys: 1,338 μd/d | Isoflavone intake[a] Girls: 13 μd/d Boys: 13μd/d | Girls: 6.8 ys ± 1.1 Boys: 9.0 ys ± 0.8 | Girls: 7.4 ys ± 0.9 Boys: 8.8 ys ± 1.0 | **Age 7–9 years** Girls: 15.9 ± 1.6 Boys: 16.4 ± 1.6 | **Age 7–9 years** Girls: 16.3 ± 2.1 Boys: 16.8 ± 2.4. | Unexposed girls were younger at baseline, and they were more likely to be from a smoking household, consume less fruit fiber, and have a lower baseline energy intake. Boys in exposure group consumed less total fiber. |
| **Duitama 2018** | Colombia | Healthy boys and girls, between 7 and 9 years old, from public schools and attending community meal locations. | 51 | 45 g of soy protein supplement dissolved in fruit juice, Monday to Saturday for 12 months | No intervention | Girls: 8.3 ys ±0.8 Boys: 8.8 ys ±0.7 | Girls: 8.2 ys ±0.9 Boys: 8.4 ys ±0.8 | **Age 7–9 years** Girls: 16.2 ± 1.2 Boys: 16.6 ± 1.2 | **Age 7–9 years** Girls: 16.4 ± 1.3 Boys: 17.4 ± 1.5 | - |
| **Giampietro 2004** | Italy | Exposure: children in prevention or treatment of cow's milk allergy. Unexposed: healthy children | 66 | Exclusive feeding of soy-based formulas for at least 6 months of life (150 kcal / kg) | No feeding of soy-based | 3.1 ys ±2 | 3.8 ys ±1.9 | - | - | Height and weight, as BMI, were within the normal range compared with those of children of the same age, sex, and race. |
| **Segovia-Siapco 2014** | United State | Girls ages 12 to 18 years attending middle and high schools near two Seventh-day Adventist universities in in California and Michigan | 131 | Soy consumption >3×/day | Soy consumption < 1x/ week | 14.9 ys ±1.5 | 15.2 ys ±1.8 | 21.5 (3.8) [b] | 22.3 (3.7) [b] | There was a significant difference between the groups in terms of site of study (California vs Michigan), type of school (public vs private) and exposure participants were more likely to have parents with a graduate-level education. |

*(Continued)*

**Table 1.** (Continued)

| Author | Country | Study population | N° | Infant diet | | Mean (SD) age at baseline | | Mean (SD) BMI | | Confounders |
|---|---|---|---|---|---|---|---|---|---|---|
| | | | | Exposure | Unexposed | Exposure | Unexposed | Exposure | Unexposed | |
| **Segovia-Siapco 2017** | United State | Boys aged 12–18 years attending middle and high schools near two Seventh-day Adventist universities in in California and Michigan | 150 | Isoflavone intake > 20 mg/day | Isoflavone intake < 3 mg/day | 14.9 ys ±1.7 | 15.0 ys±1.8 | 0.07±0.94 [c] | 0.36±1.05 [c] | There was a significant difference between the groups in terms of site of study (California vs Michigan), type of school (public vs private) and exposure participants were more likely to have parents with a graduate-level education. |
| **Sinai 2018** | Israel | Newborns born during 2004–2006 were followed up through telephone for assessing the development of milk allergy during the first 6 months of life calls, and for milk formula intake until age 3 years. | 89 | Soy-based formula for more than 3 months | Not receiving soy formula. | 8.92 ys (8.21, 9.42)[d] | 8.99 ys (8.35, 9.42)[d] | 0.67 ± 1.01[c] | 0.53 ± 1.02[c] | There were no significant differences between groups with respect to age, birthweight, gender distribution, maternal characteristics, weight for gestational age, family history, or behavioral habits. Children who had early signs of puberty reported less weekly physical activity compared to those with no pubertal signs. |
| **Strom 2001** | United State | Mostly white healthy adults, who as children participated in a controlled observational study (cow's milk versus soy milk) | 811 | Milk (formula) classified as soy based | Cow's milk | Adults aged 20 to 34 ys | Adults aged 20 to 34 ys | Women: 22.8 ±3.3 Men: 25.6 ±4.6 | Women: 22.9 ±3.7 Men: 24.8 ±3.6 | - |

UK: United Kingdom; N°: Children followed in each study,—no information provided

[a] mean

[b] median BMI (IQR)

[c] Mean BMI-for-age *z* scores (SD)

[d] Median and IQR.

**Table 2. Outcome results.**

| Author year | Follow-up | N° of PP | ♀ Age at onset of thelarche time (n/mean/SD) | ♂ Age at onset of pubarche time (mean/SD) | ♀ Age at menarche (mean/SD or median/IQR) | ♂ Age at onset of testicular growth (n/mean/SD) | ♂ Age at onset of Voice Change (n/mean/SD) | Age at first ejaculation n/mean/SD | Height in the last visit |
|---|---|---|---|---|---|---|---|---|---|
| **Adgent 2012** | Age at menarche was assessed through questionnaires annually. between ages 8 and 14.5 | - | - | - | E (54): 12.4 ys [IQR, 11.6–13.3]. Une (2124): 12.8 ys [IQR, 12–13.6] | - | - | - | - |
| **Cheng 2010** | 3 to 6 months until adulthood | - | E (40):10.7 ys ± 1 Une (39):9.9 ys ± 1.2 | - | E (40): 13.1 ys ± 1.2 Une (39):12.6 ys ± 1.0 | E (36):10.8 ys ± 0.9 Une (36):11.2 ys ± 1.0 | E (36):13.8 ys ± 1.0 Une (36):13.5 ys ± 1.2 | - | - |
| **Duitama 2018** | 12 months | E (29):0 Une (22): 0 | - | - | - | - | - | - | Mean height (SD) in cm at baseline: **E:** 127 ± 1 **Une:** 124 ± 1 Mean height (SD) in cm 12 months after exposure: **E—boys:** 130 cm ± 0.05/girls: 125 cm ± 0.07 **Une -b**oys: 124 cm ± 0.04/girls: 122 cm ± 0.06 |
| **Giampietro 2004** | 12 months | E (48):0 Une (18): 0 | - | - | - | - | - | - | - |
| **Segovia-Siapco 2014** | Cross-sectional | - | - | - | E (69):12.6 ys ± 1.3 Une (62):12.5 ys ± 1.4 | - | - | - | - |
| **Segovia-Siapco 2017** | Cross-sectional | - | - | ♂ E (81): 12.5 ys ± 0.768 ♂ Une (69): 13 ys ± 0.87 | - | - | - | - | - |
| **Sinai 2018** | 7.8 and 10.5 years | E (29):1 Une (60): 4 | - | - | - | - | - | - | Mean height $z$ scores at beginning of puberty (±9 years): **E:** - 0.17 ± 1.08/ **Une:** - 0.16 ± 1.01 |
| **Strom 2001** | 1965–1978 | - | E (128): 12.6 ys ± 1.4 Une (267): 12.7 ys ± 1.3 | ♂ E (115): 13.9 ys ± 1.6 ♂ Une (286): 13.7 ys ± 1.7 | E (128): 12.6 ys ± 1.4 Une (267): 12.7 ys ± 1.3 | - | E (111): 14.3 ys ± 1.7 Une (262): 14.0 ys ± 1.5 | E (108): 13.2 ys ± 1.2 Une (274): 13 ys ± 1.4 | Mean height (SD) in inches at adult height. **Men** E: 71.7 ± 2.5; Une: 71.4 ± 2.6 **Women** E: 65.3 ± 2.3; Une: 65.5 ± (2.5) |

PP: Precocious puberty; E: Exposure; Une: Unexposed; BMI: Body mass index; ATO: Age at take-off; PHV: Peak height velocity; IQR: Interquartile range;—no information provided.

**Table 3. Newcastle–Ottawa Scale for observational studies.**

| Author (year) | Selection | Comparability | Outcome | TOTAL |
|---|---|---|---|---|
| Adgent (2012) | 4/4 | 2/2 | 1/3 | **7/9** |
| Cheng 2010 | 2/4 | 2/2 | 1/3 | **5/9** |
| Giampietro (2004) | 2/4 | 2/2 | 2/3 | **6/9** |
| Segovia-Siapco (2014) | 3/4 | 2/2 | 2/3 | **7/9** |
| Segovia- Siapco (2017) | 3/4 | 2/2 | 2/3 | **7/9** |
| Sinai (2018) | 3/4 | 2/2 | 2/3 | **7/9** |
| Strom (2001) | 3/4 | 2/2 | 2/3 | **7/9** |

did not show an association between exposure and height at the last follow-up (Fig 4). However, in the Duitama et al. study, when comparisons were made between groups in SD and by gender, there was a statistically significant difference between exposed and unexposed girls (-0.11 vs -0.8, $p$ = 0.016). We could not plot the data from the Giampietro et al. study and the Sinai et al. study in the meta-analysis (the height was presented only by figures and in SD, respectively). In the first study, the authors reported that the children's height was within the normal range, and in the second study, there was no difference in SDs between the groups (MD 0.01 scores, 95% CI -0.46 to 0.48, 89 participants).

Investigation of publication bias was not possible owing to the small number of included studies (<10) [40].

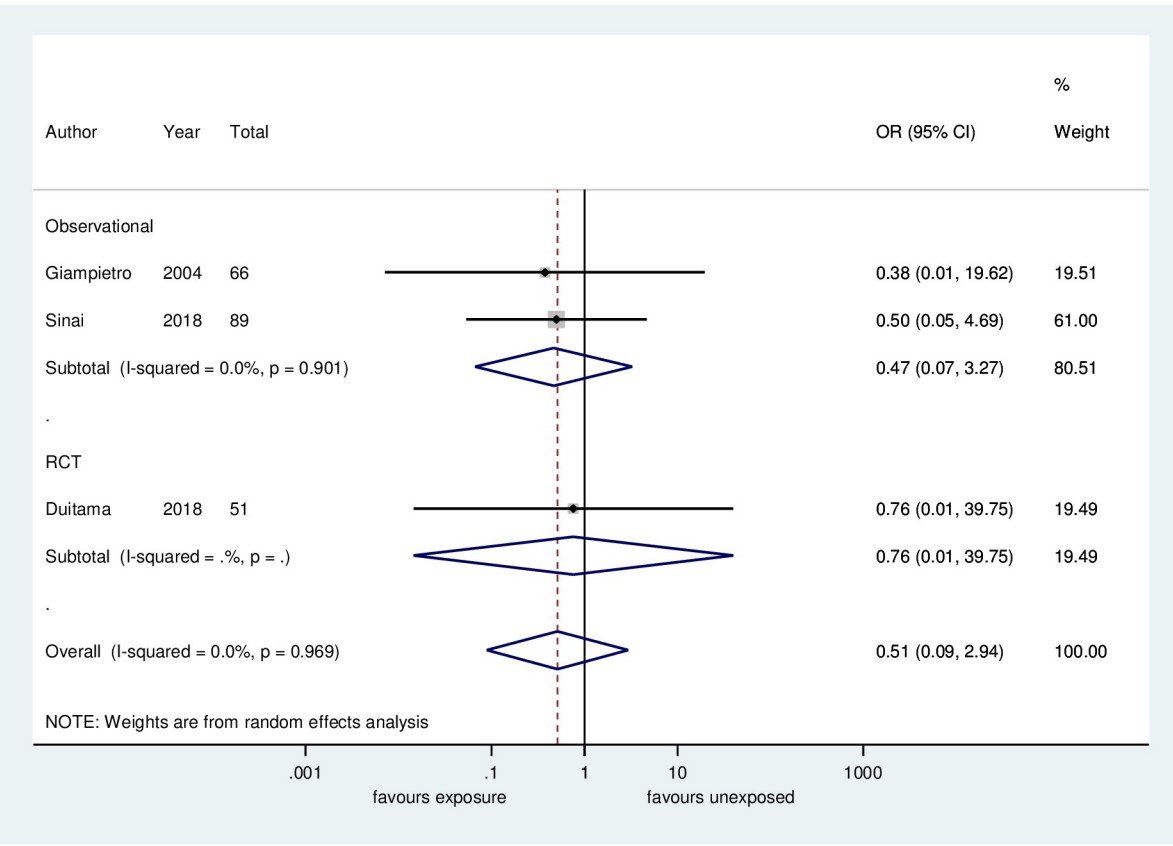

**Fig 2. Meta-analysis of the frequency of precocious puberty: Soy-based diet versus non-soy-based diet.**

**Table 4. Summary of findings–quality of evidence according to GRADE approach of association between soy-based infant diet and the onset of puberty.**

| N° of Participants (studies) Follow up | Risk of bias | Inconsistency | Indirectness | Imprecision | Publication bias | Overall certainty of evidence | Study event rates (%) | | Odds ratio (95% CI) | Anticipated absolute effects | |
| | | | | | | | Non-soy-based infant diet | Soy-based infant diet | | Effect with non-soy-based infant diet | Effect size with soy-based infant diet |
| --- | --- | --- | --- | --- | --- | --- | --- | --- | --- | --- | --- |
| **Menarche Age** | | | | | | | | | | | |
| 605 (3 studies) Childhood to Adulthood | serious [a] | not serious | not serious | serious [b] | none [c] | ⊕⊕◯◯ LOW [d,e] | - | - | - | The mean menarche age was 12.6 years | **MD 0.14 years higher** (95% CI from 0.16 lower to 0.45 higher) |
| **Precocious Puberty** | | | | | | | | | | | |
| 206 (3 studies) 1 to 2 years | serious [a] | not serious | not serious | serious [b] | none [c] | ⊕⊕◯◯ LOW [d,e] | 4/100 (4.0%) | 1/106 (0.9%) | 0.52 (0.09 to 2.94) | The risk of PP was 4 per 100 | **Risk difference was 3 fewer per 100** (95% CI from 7% less to 1% more) |
| **Height (centimeters)** | | | | | | | | | | | |
| 862 (2 studies) Childhood to Adulthood/ 1 year | serious [a] | not serious | not serious | serious [b] | none [c] | ⊕⊕◯◯ LOW [d,e] | - | - | - | The mean height was 151 cm | **MD 0.43 cm higher** (95% CI from 1 cm lower to 1.9 cm higher) |
| **Height (z score)** | | | | | | | | | | | |
| 89 (1 study) 7.8 to 10.5 years | serious [a] | not serious | not serious | serious [b] | none [c] | ⊕⊕◯◯ LOW [d,e] | - | - | - | The mean z score of height was 0.16 | **MD 0.01 score lower** (95% CI from 0.46 lower to 0.48 higher) |

**GRADE:** Grading of Recommendations Assessment, Development, and Evaluation; **CI:** Confidence interval; **MD:** Mean difference; **PP:** Precocious puberty; **RCT:** Randomized controlled trial

*Explanations*

a. In most observational studies the representativeness of the exposed cohort was a selected group. Most studies reported loss of follow- up, and data on outcomes which prevented the assessment of the participants. Pubertal development was self-reported in four studies. In the RCT there is no information regarding randomization process, allocation concealment and attrition bias.

b. All meta-analyses presented a small sample size, and the optimal information size was not achieved, the effect size crossed the line of no effect, and confidence interval was wide.

c. Investigation of publication bias was not possible due to the small number of included studies (<10). However, as we performed a huge search strategy comprising published and unpublished studies, we did not consider rating down the quality of evidence in this domain.

d. Most of the patients included were from observational studies that did not meet any of the criteria that could increase the certainty of evidence (such as the large magnitude of the treatment effect, the dose-response relationship, the plausible biases decreasing the magnitude of the exposure effect).

e. Low certainty of evidence: Since our confidence in the effect estimate is limited, the true effect may be substantially different.

**Studies included in the qualitative synthesis.** Only the Strom et al. study [17] assessed the age at first ejaculation, and no significant differences were found between the groups. Testicular growth was analyzed only by Cheng et al. [15], and no association between exposure and this outcome was found. Segovia-Siapco et al. [16] and Strom et al. [17] evaluated pubarche in boys. Segovia-Siapco et al. [33] reported that the mean age of pubarche was 12.5 years in exposed and 13 in unexposed (MD -0.5 years, 95% CI -0.77 to -0.23). Strom et al. [17] reported that the mean age was 13.9 in the exposed group and 13.7 years in the unexposed group (MD 0.2, 95% CI -0.15 to 0.55).

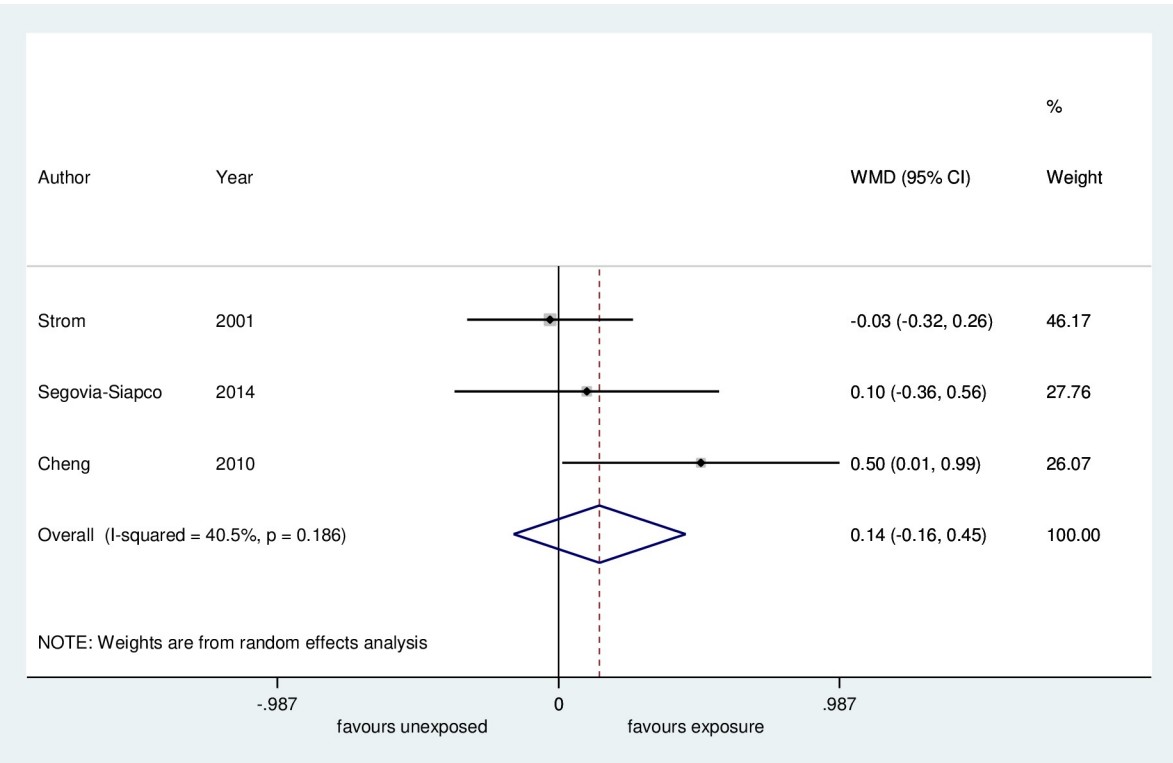

**Fig 3. Meta-analysis of mean age of menarche: Soy-based diet versus non-soy-based diet.** WMD: Unstandardized mean difference.

Cheng et al. [15] and Strom et al. [17] evaluated the age of thelarche. In the first study, the mean age for this outcome was 10.7 years for the exposed and 9.9 for the unexposed, with a mean difference of 0.8 years (95% CI 0.31 to 1.29, 79 patients). In the second study, mean age was 12.3 years in both groups (MD -0.01; 95% CI -0.33 to 0.32; 395 patients). The heterogeneity between these two studies can be explained by the criteria used for thelarche. Cheng et al. considered it to be self-reported Tanner stage 2 for breast development, while Strom et al. considered it to be when the breast developed sufficiently to begin wearing a bra.

No study assessed the risk of delayed puberty, as well as the age at onset of pubarche in girls.

## Discussion

In the current review, we evaluated the association between a soy-based infant diet and the onset of puberty in girls and boys. Eight studies were included, with a total of 598 children exposed to a soy-based diet, and 2957 were unexposed. The outcomes that were plotted in the meta-analyses were the risk of precocious puberty, age at menarche, and height at the last follow-up. We did not find any association between exposure and these outcomes. In one study, the age at thelarche was marginally increased in girls with a soy-based infant formula. However, the reported age was within the normal age range for menarche [38]. Segovia-Siapco et al. [16] also reported that a high total soy isoflavone intake was significantly associated with earlier median age pubarche in boys. Cheng et al. reported that girls whose diet was in the highest dietary isoflavone tertile experienced Tanner stage 2 for breast development later than did girls whose diet was in the lowest isoflavone tertile [15]. However, in both situations, the reported age was within the normal age range for pubarche in boys and thelarche in girls.

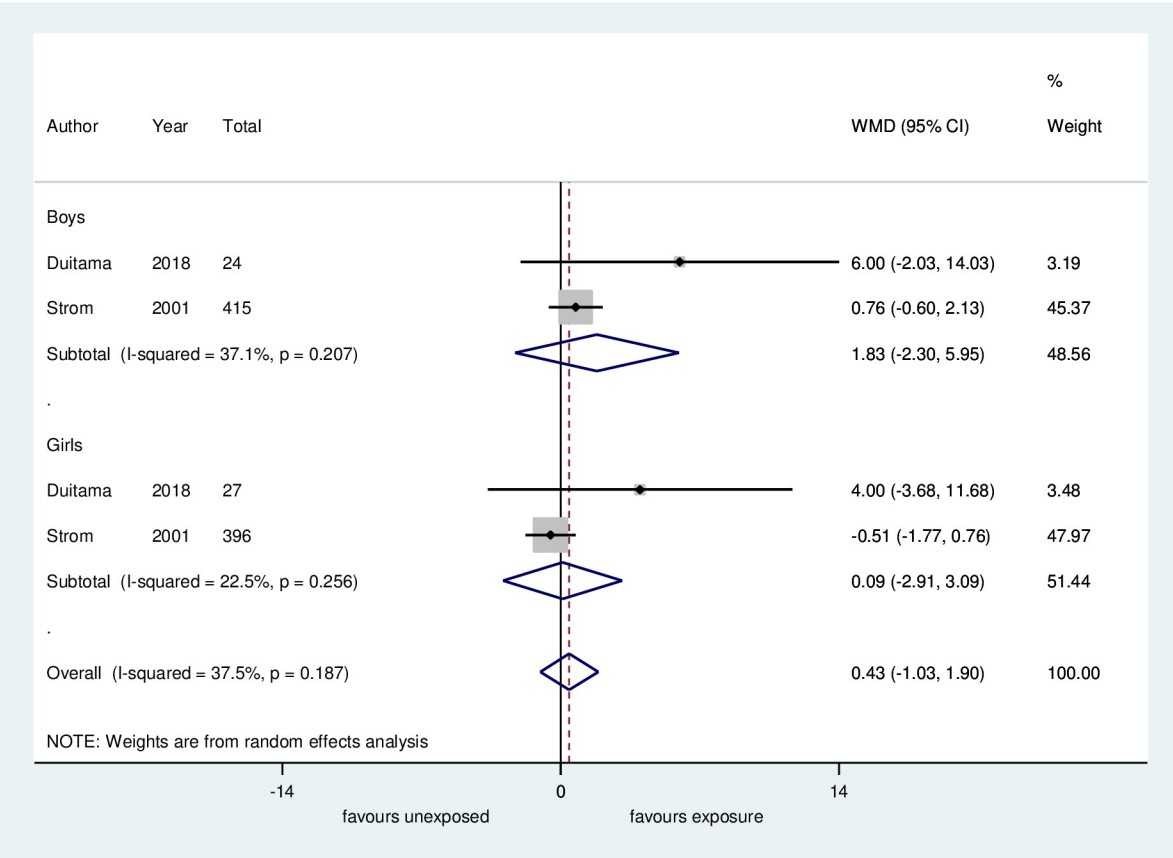

**Fig 4. Meta-analysis of height in the final follow-up: Soy-based diet versus non-soy-based diet.** WMD: Unstandardized mean difference. In the Duitama study height was measured at the beginning of puberty, and in the Strom study height was measured as an adult.

This is the first systematic review to evaluate the association between a soy-based infant diet and the onset of puberty in girls and boys. The controlled studies (exposed vs. unexposed) on this topic were included in this review. Two Korean case–control studies have also evaluated the association between soy and puberty onset. In the first study, Kim et al. compared serum isoflavone concentrations in 101 girls with idiopathic central precocious puberty with 91 age-matched controls [41]. Serum concentrations of daidzein, genistein, and total isoflavones were significantly higher in children with precocious puberty than in normal children. In the second study, equol, genistein, and daidzein were measured in the plasma of 150 girls with precocious puberty and compared with 90 control subjects. Diadzein and equol levels in the plasma of precocious puberty patients were respectively 1.37 and 1.3 times higher than those of the control group. However, there was no significant difference. Genistein was significantly 2.67 times higher in girls with precocious puberty than in normal girls [42]. Thus, in both studies, the authors concluded that high serum isoflavones may be associated with the risk of precocious puberty in Korea. The important point that distinguishes the two case–control studies from the studies included in our review is that we compared the risk of precocious puberty and the age at the onset of puberty in participants who were exposed to a soy-based infant diet. Conversely, in the two Korean studies, children with central precocious puberty were compared with normal children in terms of isoflavone serum concentrations. After a clinical observation, the first step to explore the association of causality between exposure and outcome is to perform a case-control study [43]. It is an efficient method of investigation when a disease

occurrence is rare. However, there were important limitations inherent in the study design. For example, in a case–control study, as exposure is assessed after the outcome, it can be difficult to establish if it was preceded by exposure. As in cohort studies, exposure precedes the disease, if a case–control study shows evidence that a certain exposure is suspected, the next step should be to carry out a cohort study [43]. The cohort studies included in our review did not confirm this association in the Korean studies.

However, our systematic review has some limitations. The main limitation is related to the small number of studies that evaluated the risk of precocious puberty. Additionally, two of the three studies that assessed this outcome were studies with no events in either arm. These studies were naturally excluded from meta-analyses of OR and relative risk [24]. However, there was no consensus regarding whether these studies should be included. Xu et al. showed that excluding such studies may influence the magnitude and direction of the treatment effects [44]. Cheng et al. recommend including studies with no events in both arms in meta-analyses when treatment effects are unlikely, although they should be excluded when there is treatment effect [23]. In our review, we performed meta-analyses including and excluding these studies. In both analyses, there was no statistical difference between the groups in the risk of precocious puberty. Second, because of the small number of studies included in the meta-analyses, we could not perform sensitivity analyses. The third limitation was that, in most studies, the exact amount of soy consumption was not reported. Fourth, our results were predominantly from retrospective studies that did not meet any of the criteria that could increase the certainty of evidence. Additionally, for children who consumed soy-based milk, only Sinai et al. and Giampietro et al. reported that the exposure group consisted of patients in prevention or with an allergy to cow's milk. Although most studies reported that participants were from healthy populations, many of the children could have had a condition that forced the parents to use soy milk instead of cow's milk. We could not evaluate whether this condition could interfere with the association between soy and the onset of puberty. The fifth limitation was that in most studies included reports of puberty stage, progression of puberty, the expected mid-parental height and parental puberty were not assessed by a physician.

## Conclusion

### Implications for practice

We did not observe any association between a soy-based infant diet and the onset of puberty in boys or girls. The meta-analyses did not show any significant differences between groups in the risk of precocious puberty and menarche age. Individual studies have not shown significant clinical differences between groups regarding menarche age, age of testicular enlargement, pubarche, and voice change in boys. No study assessed the risk of delayed puberty as well as the age at onset of pubarche in girls.

### Implications for research

Cohort studies with large sample sizes and appropriate methodology are required to evaluate whether the lack of an association between a soy-based infant diet and the onset of puberty remains when the results are compared considering the length of exposure, amount of soy consumption, and assessment of pubertal development through self-report or physical examination.

## Supporting information

**S1 Fig. Figure of meta-analysis of the frequency of precocious puberty (excluding the studies with no events in both arms).**
(DOCX)

**S1 File. Search strategy.**
(DOCX)

**S2 File. PRISMA checklist.**
(DOCX)

## Acknowledgments

We thank Dr. Lehana Thabane for providing references on how to evaluate studies with no events in both arms in the meta-analyses.

## Author Contributions

**Conceptualization:** Ana Flora Silva e Gustavo, Renan Braga Gonçalves, Vania dos Santos Nunes-Nogueira.

**Data curation:** Flávia Ramos Kazan Oliveira, Ana Flora Silva e Gustavo, Fernanda Bolfi, Vania dos Santos Nunes-Nogueira.

**Formal analysis:** Flávia Ramos Kazan Oliveira, Ana Flora Silva e Gustavo, Renan Braga Gonçalves, Fernanda Bolfi, Vania dos Santos Nunes-Nogueira.

**Funding acquisition:** Vania dos Santos Nunes-Nogueira.

**Investigation:** Renan Braga Gonçalves, Vania dos Santos Nunes-Nogueira.

**Methodology:** Ana Flora Silva e Gustavo, Renan Braga Gonçalves, Fernanda Bolfi, Vania dos Santos Nunes-Nogueira.

**Project administration:** Vania dos Santos Nunes-Nogueira.

**Resources:** Vania dos Santos Nunes-Nogueira.

**Software:** Ana Flora Silva e Gustavo, Renan Braga Gonçalves, Vania dos Santos Nunes-Nogueira.

**Supervision:** Vania dos Santos Nunes-Nogueira.

**Validation:** Vania dos Santos Nunes-Nogueira.

**Visualization:** Adriana Lúcia Mendes, Vania dos Santos Nunes-Nogueira.

**Writing – original draft:** Flávia Ramos Kazan Oliveira, Adriana Lúcia Mendes, Vania dos Santos Nunes-Nogueira.

**Writing – review & editing:** Vania dos Santos Nunes-Nogueira.

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
