## [Decision Letter · Decision Letter 0]

25 Sep 2020

PONE-D-20-20990

Effect of Soy-Based Diet in the Timing of Puberty: A Systematic Review and Meta-Analysis

PLOS ONE

Dear Dr. Nunes,

Thank you for submitting your manuscript to PLOS ONE. After careful consideration, we feel that it has merit but does not fully meet PLOS ONE’s publication criteria as it currently stands. Therefore, we invite you to submit a revised version of the manuscript that addresses the points raised during the review process.

Your manuscript has been reviewed by two experts in the field, and they have found some points that need to be addressed before this manuscript is considered for publication. Please go through the reviewers' comments and consider addressing these points, and prepare a revised version.

We look forward to receiving your revised manuscript.

Kind regards,

Ivan D. Florez

Academic Editor

PLOS ONE

Additional Editor Comments:

Your manuscript has been reviewed by two experts in the field, and they have found some points that need to be addressed before this manuscript is considered for publication. Please go through the reviewers' comments and consider addressing these points, and prepare a revised version.

Journal Requirements:

2. Please address the following points:

1) We note that publication bias has not been assessed. Please provide an assessment using both graphs (funnel plots) and statistical methods

2) Please revise your introduction to ensure that all statements are supported by appropriate references.  Moreover, we note that reference 10 refers to a study conducted on animals, not on human participants; please revise the statement made in the introduction referring to this citation.

3)  In your Abstract, please consider including a statement regarding the overall quality  of evidence.

4) Please consider reporting the full results of your quality assessment in the main text, and not in the Supplementary file.

- https://academic.oup.com/edrv/article/30/4/293/2355049

In your revision ensure you cite all your sources (including your own works), and quote or rephrase any duplicated text outside the methods section. Further consideration is dependent on these concerns being addressed.

5. We note that this manuscript is a systematic review or meta-analysis; our author guidelines therefore require that you use PRISMA guidance to help improve reporting quality of this type of study. Please upload copies of the completed PRISMA checklist as Supporting Information with a file name “PRISMA checklist”.

Reviewers' comments:

Reviewer's Responses to Questions

**Comments to the Author**

1. Is the manuscript technically sound, and do the data support the conclusions?

Reviewer #1: Partly

Reviewer #2: Partly

2. Has the statistical analysis been performed appropriately and rigorously? 

Reviewer #1: No

Reviewer #2: I Don't Know

3. Have the authors made all data underlying the findings in their manuscript fully available?

Reviewer #1: Yes

Reviewer #2: Yes

4. Is the manuscript presented in an intelligible fashion and written in standard English?

Reviewer #1: Yes

Reviewer #2: Yes

5. Review Comments to the Author

Reviewer #1: The authors reviewed an extremely important topic that is highly debatable among pediatricians and pediatric endocrinologist. I have comments that are important to improve the reporting and end-user utilization:

- please add a reference to this statement is not "Efforts to implement healthier eating habits have resulted in an increase in the consumption of 47 soy-based products."

- the definition of population need to be more specific: were children with chronic illness excluded, did you included children from all ethnic background

- what did you consider an intake above the population mean for soy product?

- line 114: please correct the age for delayed puberty for girls is 13, boys 14

- do you mean final adult height? Or height assessed at the beginning of pubertal signs?

- how was the pubertal assessment done? was it assessed by a physician or self-reported

- in the statistical analysis section the authors need to elaborate on the method used to analyze the frequency of precocious puberty. Please see comment below regarding analysis of zero events rate.

- please describe GRADE evaluation in brief for the readers

-In the results please give details of the exposure in the included studies e.g dose, length of exposure, product type, the starting age of follow up

-Line 200-2001 Please clarify the sentence “ the exposed cohort was a selected group, and not as a representative of the average population in the community type of selected”

-In the meta-analysis, please do appropriate statistical correction for analyzing count data that include zero events. The current analysis model is not appropriate.

- Why the age of thelarche, pubarche, voice change were not meta-analyzed?

-For the height outcome, it is hard to interpret not knowing if height is assessed at similar pubertal stage or final adult height. However, in the figure the interpretation of data can be simplified for clinicians by changing the label of the figure to something like” taller with soy” vs. “taller with control”. It seems that the children consuming Soy are taller than controls, although the CI is touching zero. This needs to be heightened in the results & discussion.

From this data its suspicious that those children had gone through puberty earlier.

- please do sensitivity analysis to compare results of length of exposure to soy product, if puberty was assessed vs. self-reported, boys vs. girls, cohort year (because of the secular trend in pubertal achievement).

- table 1 in the supplementary material should be moved to the manuscript

- why heterogeneity was not explored?

-Please organize the authors in table 1 in alphabetical order, include the length of follow up, type of population (healthy, children with CMP allergy), amount of consumed soy products per day, ethnic group as separate column. Can you include possible effect modifers

-There are a list of outcomes planned in the methods section to be reported with no results reported. Can you please provide data on these outcomes.

-The results were not expected, can you please provide possible explanations. Also, can you please compare the intake reported in the included studies to the typically reported to cause precocious puberty

- your review has a unique opportunity to report on methods used to report puberty assessment. Please add a paragraph in the discussion to discuss the appropriateness of the used methods in the included studies and ways that could improve future research report.

-Please discuss gaps to be addressed in future research

- the conclusion statement need to be re-phrased to capture the evidence quality

Reviewer #2: I appreciate the effort of the authors in gathering collective evidence on the status of soy intake in relation to pubertal onset. The supplementary material indicates that there had been a thorough search of literature for articles that could potentially be included in the review. It seems that the authors also made sure to test the suitability of conducting a meta-analysis. However, I have the following concerns:

(1) The aim of the study, stated as “evaluate the EFFECT of exposure to a soy-based infant feeding or to a soy-rich diet during childhood on the timing of onset of puberty in girls and boys” was obviously not consistent with the choice of articles included in the review. One can only determine “effect” in the context of a well-designed clinical trial/experimental study, particularly a randomized controlled trial.

(2) Observational studies, where comparisons between the exposed group (ie, consumed soy/soy constituents) and an unexposed group (i.e., those that didn’t consume soy/soy constituents ), had been repeatedly called “controlled” studies. Considering that there were only 8 studies, and all but one (an RCT) are observational studies, this is misleading. A more appropriate term should be used to be more consistent with the types of articles that were included in the review.

(3) Although understandable, several outcome variables were considered for meta-analysis even if in some cases, only 1 or 2 studies was/were the source of “evidence” on which to base conclusions for a particular outcome. The heterogeneity tests all turned out non-significant possibly due to this (although already expected since the findings from the articles are not dissimilar). Very few studies are currently existing on this topic and this limits the ability to extract a well-informed or solid evidence on associations between soy intake and timing of puberty. Based on the existing evidence, however, can something conclusive be determined? If so, why is that so?

Here are additional comments/feedback and questions on the work:

1. Abstract:

lines 20-22 --Often, we measure the risk (of precocious puberty, in this case); why was "frequency" used and what does it mean? Did you mean mean this to be "count" (i.e., number of children who have precocious puberty)? THis is not clear.

2. Introduction:

lines 63-64 -- The study you cited is an animal study (on rats). We don't call female animals "women".

lines 63-64: The statement is not clear --What substances and what receptors?

line 73: Are you sure about the unit for this value (423.4 mg/day? The article says microgram/day (ug/day).

line 76: median intake is 10 milligrams/day (mg/day), as indicated in that study.

lines 84-86: "Effect" is the main aim of the review. However, the studies included in the review are mostly observational studies except for 1 RCT. Exposure mentioned here is "soy" but articles chosen included soy constituents (specifically the isoflavones). This aim needs rewriting.

3. Methods:

Criteria for eligibility section -- This part is a bit confusing because I'm expecting this to be focused on your criteria for selection of articles, but in some parts of this section, you are referring to individuals/study participants instead of the studies. Your focus should be on the articles and not study participants/individuals.

lines 103-104: What "population" serves as the reference then? Did you consider this as your "cut-off" for soy intake? Isn't it that exposure or non-exposure to soy in its different forms (soy foods, products, formula) and/or its constituents (soy isoflavones) regardless of amount eaten is your independent /exposure variable?

Lines 111-117: There are so many outcomes listed here and they appear to be different or not even related to each other. In your aim, you stated that you want to determine how soy exposure  timing of puberty. I suggest that you stick to that and then state under this section what you consider to be measurements of timing of puberty before you list all that you have in the section (ie, early (such as in precocious puberty) or delayed onset of secondary sexual characteristics, in both males (pubarche, testicular growth, penile enlargement, etc.) and females (menarche, pubarche, thelarche) and other indicators of puberty (e.g., first ejaculation, voice change, growth spurt, etc.). This way, it would be clear to the reader that these "outcomes" you listed are all related to the timing of puberty. Also, why did you not consider separating the pubertal timing in boys and the pubertal timing in girls separately in your meta-analysis?

Lines 120-122: Did you mean studies without a "comparison group"? Please note that "uncontrolled studies" imply a clinical or experimental study design. Even if there is a comparison group, the exposed group cannot be considered to be a "controlled" comparison grp if the study design is not experimental.

Line 131: Please specify the databases you used for these types of studies (PhD/master's theses).

Line 134: Give a few examples of your search terms (you cannot expect readers to go to your supplementary file just for this)

Line 150: Did you mean randomized controlled trial?

4. Results:

In general, this section needs to be rewritten in order to be consistent with the other needed revisions (e.g., revised aim, etc.)

Line 211: Onset of menarche? timing of menarche

Line 219: Growth Spurt?

5. Discussion:

line 258: "controlled studies" would not be in the context of an experimental design. Please use a more appropriate term.

6. Conclusion: Considering that most of the evidence is from observational studies, there is a need to revise your statement here ("negatively alter" implies effect which is consistent with your current aim, but not appropriate based on where you got your evidence from, ie. observational studies).

7. Tables and Figures:

Figure 1:

--"Trabalhos excluidos" need to be translated to English;

--For the box "Full papers evaluated for eligibility), n=16. However, tracked articles (42) minus 29 under Trabalhos excluidos equals 13. Please correct this.

Table 1:

--This table would be more informational if the description of the studies were more succinct and if the study findings/conclusions were added. For example, "Outcome evaluation age" is redundant since the information is already under the "Patients" column (Why do you use the term patients instead of "Study Population"?The latter is more appropriate considering the study design of the articles). The exclusion criteria is not helpful--why not add information about variables/confounders that were controlled for by the study?

Table 2:

--This table is not helpful to my understanding of the quality of evidence. Please make it more stand-alone -- what do the symbols under the column "Certainty of the Evidence" mean? Also, there is no clear explanation in the text about how you came up with the quality of evidence. You only referred the reader to the table when discussing your results.

I hope this review will be helpful to the authors.

6. PLOS authors have the option to publish the peer review history of their article (what does this mean?). If published, this will include your full peer review and any attached files.

Reviewer #1: **Yes: **Reem Al Khalifah

Reviewer #2: **Yes: **Gina Segovia-Siapco

---

## [Author Response · Author response to Decision Letter 0]

8 Dec 2020

December 07, 2020

PLOS ONE

Editorial Board

Dear Editor:

Thank you very much for your email dated 09-Sep-2020 with reviews comments on our manuscript “The association between soy-based infant diet and the onset of puberty: A systematic review and meta-analysis”

The reviewers’ comments were very helpful. We thank you for the opportunity to address the comments in a revised version of the manuscript, and therefore we have revised the manuscript in accordance with the feedback.

On behalf of my co-authors, I am submitting the manuscript with the requested revision.

We hope that we have adequately addressed all the comments, and that this version is now acceptable for publication in the prestigious Plos One

Below are our responses to the specific issues raised in the comments:

Journal Requirements:

1. Please ensure that your manuscript meets PLOS ONE's style requirements, including those for file naming. The PLOSONE style templates can be found at

R.: We have reviewed and made corrections

2. Please address the following points:

1) We note that publication bias has not been assessed. Please provide an assessment using both graphs (funnel plots) and statistical methods 

R. We have not assessed publication bias either using graphs or statistical methods due to the number of studies (<10). We have emphasized this in the manuscript, line 243-244 and 324-325

2) Please revise your introduction to ensure that all statements are supported by appropriate references. Moreover, we note that reference 10 refers to a study conducted on animals, not on human participants; please revise the statement made in the introduction referring to this citation. 

R. We have revised and corrected all introduction. “Experimental animal studies have showed that xenobiotics as polychlorinated biphenyls (PCBs), phytoestrogens, fungicides, pesticides may affect sexual differentiation [9]. This interference has a high probability of affecting both reproductive physiology and behavior at several stages of life [10]. In female rats, an early exposure (late embryonic and / or early postnatal) to low doses of PCBs [10] or soy significantly affected mating behavior”

3) In your Abstract, please consider including a statement regarding the overall quality of evidence.

R. We have included; line 39-40

 4) Please consider reporting the full results of your quality assessment in the main text, and not in the Supplementary file. 

R. We have reported the full results of quality assessment in the main text (Table 2 and 3)

3. We noticed you have some minor occurrence of overlapping text with the following previous publication(s), whichneeds to be addressed:

- https://academic.oup.com/edrv/article/30/4/293/2355049

In your revision ensure you cite all your sources (including your own works), and quote or rephrase any duplicated text out side the methods section. Further consideration is dependent on these concerns being addressed. 

R. We have revised all introduction in order to avoid this occurrence

4. Please include captions for your Supporting Information files at the end of your manuscript, and update any in-textcitations to match accordingly. Please see our Supporting Information guidelines for more 

R. We have included

5. We note that this manuscript is a systematic review or meta-analysis; our author guidelines therefore require that youuse PRISMA guidance to help improve reporting quality of this type of study. Please upload copies of the completed PRISMA checklist as Supporting Information with a file name “PRISMA checklist”.

Reviewers' comments: Reviewer's Responses to Questions

R. We have completed (S3_File)

1. Is the manuscript technically sound, and do the data support the conclusions? The manuscript must describe a technically sound piece of scientific research with data that supports the conclusions. Experiments must have been conducted rigorously, with appropriate controls, replication, and sample sizes. The conclusions must be drawn appropriately based on the data presented.

Reviewer #1: Partly

Reviewer #2: Partly

R. We have revised and updated the introduction, discussion and conclusion

2. Has the statistical analysis been performed appropriately and rigorously?

Reviewer #1: No

Reviewer #2: I Don't Know

R. We have revised all methodology, especially the meta-analysis of precocious puberty, line 210-219

3. Have the authors made all data underlying the findings in their manuscript fully available? The PLOS Data policy requires authors to make all data underlying the findings described in their manuscript fullyavailable without restriction, with rare exception (please refer to the Data Availability Statement in the manuscript PDFfile). The data should be provided as part of the manuscript or its supporting information, or deposited to a publicrepository. For example, in addition to summary statistics, the data points behind means, medians and variance measures should be available. If there are restrictions on publicly sharing data—e.g. participant privacy or use of data from a third party—those must be specified.

Reviewer #1: Yes

Reviewer #2: Yes

4. Is the manuscript presented in an intelligible fashion and written in standard English? PLOS ONE does not copyedit accepted manuscripts, so the language in submitted articles must be clear, correct, and unambiguous. Any typographical or grammatical errors should be corrected at revision, so please note any specific errors here.

Reviewer #1: Yes

Reviewer #2: Yes

Reviewer 1

Thank you very much for your comments and suggestions that absolutely improved the quality of our review. So, we have updated the manuscript based on your comments and all your suggestions were included in the revised manuscript, as you can see below:

The authors reviewed an extremely important topic that is highly debatable among pediatricians and pediatric endocrinologist. I have comments that are important to improve the reporting and end-user utilization:

 - please add a reference to this statement is not "Efforts to implement healthier eating habits have resulted in an increase in the consumption of 47 soy-based products."

R. We have updated this phrase and include the reference, line 86-87

 - the definition of population need to be more specific: were children with chronic illness excluded, did you included children from all ethnic background 

R. We have specified, line 140-143

- what did you consider an intake above the population mean for soy product?

R. We have rewritten this phrase, line 144-149

 - line 114: please correct the age for delayed puberty for girls is 13, boys 14

R. We have corrected, line 158-159

 - do you mean final adult height? Or height assessed at the beginning of pubertal signs? 

R. We have changed this expression to height in the last follow-up, line 160

- how was the pubertal assessment done? was it assessed by a physician or self-reported 

R. We have included this information, line 287-288

- in the statistical analysis section the authors need to elaborate on the method used to analyze the frequency ofprecocious puberty. Please see comment below regarding analysis of zero events rate. 

R. We have updated the data analysis, line 210-217

- please describe GRADE evaluation in brief for the readers

R. We have included this information, line 232-242

- In the results please give details of the exposure in the included studies e.g dose, length of exposure, product type, the starting age of follow 

R. We have included this information, line 258-267

-Line 200-2001 Please clarify the sentence “ the exposed cohort was a selected group, and not as a representative of the average population in the community type of selected” 

R. We have updated all risk of bias assessment and this phrase was eliminated, line 281-290

-In the meta-analysis, please do appropriate statistical correction for analyzing count data that include zero events. The current analysis model is not appropriate.

R. We have updated the data analysis, line 210-217

- Why the age of thelarche, pubarche, voice change were not meta-analyzed?

R. They were not meta-analyzed because only one study evaluated this outcome, or two evaluated, but with heterogeneity between two results. We have included this information in the methodology, line 226 “If enough studies were available, the potential causes of heterogeneity among studies could be evaluated by subgroup analysis. In case of considerable inconsistency (I2 > 50%) associated to variation in the direction of association, and heterogeneity could not be explained, we did not perform a meta-analysis”

 - For the height outcome, it is hard to interpret not knowing if height is assessed at similar pubertal stage or final adult height. However, in the figure the interpretation of data can be simplified for clinicians by changing the label of the figure to something like” taller with soy” vs. “taller with control”. It seems that the children consuming Soy are taller than controls, although the CI is touching zero. This needs to be heightened in the results & discussion. From this data its suspicious that those children had gone through puberty earlier. 

R. Thank you for important information. First, I have double checked the results plotted in the meta-analyses, and we realized, that in one of the studies only the data for girls was plotted. So we corrected this mistake, and in the current graph we can see a non-difference between groups. Fig 4.

- please do sensitivity analysis to compare results of length of exposure to soy product, if puberty was assessed vs. self-reported, boys vs. girls, cohort year (because of the secular trend in pubertal achievement).

R. Unfortunately, we could not be performed sensitivity analyses due to small number of studies include, and have explained this in the main manuscript, line 219-222, and in the limitations paragraph.

 - table 1 in the supplementary material should be moved to the manuscript - why heterogeneity was not explored? -Please organize the authors in table 1 in alphabetical order, include the length of follow up, type of population (healthy,children with CMP allergy), amount of consumed soy products per day, ethnic group as separate column. Can you include possible effect modifers 

R. We have updated the table 1 and included all your suggestions, and we moved for it all information that was in the supplementary data. We have explored heterogeneity in the item “Studies included in the qualitative synthesis”, line 327 

-There are a list of outcomes planned in the methods section to be reported with no results reported. Can you please provide data on these outcomes. 

R. We have updated this infomration and justified that No study assessed the risk of delayed puberty, as well as the age of onset of pubarche in girls, line 340

-The results were not expected, can you please provide possible explanations. Also, can you please compare the intake reported in the included studies to the typically reported to cause precocious puberty 

R. We have provided this in the discussion section, line 357-379

- your review has a unique opportunity to report on methods used to report puberty assessment. Please add a paragraph in the discussion to discuss the appropriateness of the used methods in the included studies and ways that could improve future research report.

R. We have included this information, line 404-407

 -Please discuss gaps to be addressed in future research

R. We have included this information, line 390-393

 - the conclusion statement need to be re-phrased to capture the evidence quality

R. We have updated this information, line 395

Reviewer 2

Thank you very much for your comments and suggestions that absolutely improved the quality of our review. So, we have updated the manuscript based on your comments and all your suggestions were included in the revised manuscript, as you can see below:

I appreciate the effort of the authors in gathering collective evidence on the status of soy intake in relation to pubertal onset. The supplementary material indicates that there had been a thorough search of literature for articles that could potentially be included in the review. It seems that the authors also made sure to test the suitability of conducting a meta-analysis. However, I have the following concerns:

 (1) The aim of the study, stated as “evaluate the EFFECT of exposure to a soy-based infant feeding or to a soy-rich diet during childhood on the timing of onset of puberty in girls and boys” was obviously not consistent with the choice of articles included in the review. One can only determine “effect” in the context of a well-designed clinical trial/experimental study, particularly a randomized controlled trial.

R. Thank you for important observation. I have updated all manuscript according to your suggestion. I have changed the term effect for association, and control for unexposed.

 (2) Observational studies, where comparisons between the exposed group (ie, consumed soy/soy constituents) and an unexposed group (i.e., those that didn’t consume soy/soy constituents), had been repeatedly called “controlled” studies. Considering that there were only 8 studies, and all but one (an RCT) are observational studies, this is misleading. A more appropriate term should be used to be more consistent with the types of articles that were included in the review.

R. Thank you for important observation. I have updated all manuscript according to your suggestion. I have changed the term effect for association, and control for unexposed.

 (3) Although understandable, several outcome variables were considered for meta-analysis even if in some cases, only 1or 2 studies was/were the source of “evidence” on which to base conclusions for a particular outcome. The heterogeneity tests all turned out non-significant possibly due to this (although already expected since the findings from the articles are not dissimilar). Very few studies are currently existing on this topic and this limits the ability to extract a well-informed or solid evidence on associations between soy intake and timing of puberty. Based on the existing evidence, however, can something conclusive be determined? If so, why is that so?

R. Thank you for important comments. I have updated all manuscript in order to change the idea of effect for association. Additionally, we have emphasized the quality of evidence according GRADE approach. The no association does not mean no effect, due to that, we think that the current version our conclusion is more apropriate. 

 Here are additional comments/feedback and questions on the work:

 1. Abstract: lines 20-22 --Often, we measure the risk (of precocious puberty, in this case); why was "frequency" used and what does itmean? Did you mean mean this to be "count" (i.e., number of children who have precocious puberty)? This is not clear. 

R. In all manuscript we have changed for risk of precocious puberty

2. Introduction: lines 63-64 -- The study you cited is an animal study (on rats). We don't call female animals "women". lines 63-64: The statement is not clear --What substances and what receptors? line 73: Are you sure about the unit for this value (423.4 mg/day? The article says microgram/day (ug/day). line 76: median intake is 10 milligrams/day (mg/day), as indicated in that study. lines 84-86: "Effect" is the main aim of the review. However, the studies included in the review are mostly observational studies except for 1 RCT. Exposure mentioned here is "soy" but articles chosen included soy constituents (specifically the isoflavones). This aim needs rewriting. 

R. We have corrected all these smistake, sorry for this, line 92, 103-106, 116-121. In line 144 we have included: The exposure group comprised individuals exposed to a soy-based infant diet. We considered as a soy-based infant diet a soy-based infant feeding (e.g., formula or milk soy-based), higher food intake of isoflavones, or the use of soy protein-based supplements

3. Methods: Criteria for eligibility section -- This part is a bit confusing because I'm expecting this to be focused on your criteria for selection of articles, but in some parts of this section, you are referring to individuals/study participants instead of the studies. Your focus should be on the articles and not study participants/individuals. lines 103-104: What "population" serves as the reference then? Did you consider this as your "cut-off" for soy intake? Isn'tit that exposure or non-exposure to soy in its different forms (soy foods, products, formula) and/or its constituents (soy isoflavones) regardless of amount eaten is your independent /exposure variable? Lines 111-117: There are so many outcomes listed here and they appear to be different or not even related to each other. In your aim, you stated that you want to determine how soy exposure  timing of puberty. I suggest that you stick to that and then state under this section what you consider to be measurements of timing of puberty before you list all that you have in the section (ie, early (such as in precocious puberty) or delayed onset of secondary sexual characteristics, in both males (pubarche, testicular growth, penile enlargement, etc.) and females (menarche, pubarche, thelarche) and other indicators of puberty (e.g., first ejaculation, voice change, growth spurt, etc.). This way, it would be clear to the reader that these "outcomes" you listed are all related to the timing of puberty. Also, why did you not consider separating the pubertal timing in boys and the pubertal timing in girls separately in your meta-analysis? Lines 120-122: Did you mean studies without a "comparison group"? Please note that "uncontrolled studies" imply a clinical or experimental study design. Even if there is a comparison group, the exposed group cannot be considered to be a "controlled" comparison grp if the study design is not experimental. Line 131: Please specify the databases you used for these types of studies (PhD/master's theses). Line 134: Give a few examples of your search terms (you cannot expect readers to go to your supplementary file just forthis) Line 150: Did you mean randomized controlled trial? 

R. We have updated the methods section, and we included all your suggestions, line 131-181. Thank you so much for all them. I have specified the two outcomes not evaluate, because the studies included did not report them, line 340.

4. Results: In general, this section needs to be rewritten in order to be consistent with the other needed revisions (e.g., revised aim,etc.) Line 211: Onset of menarche? timing of menarche Line 219: Growth Spurt?

R. We have updated the methods section, and we included all your suggestions line 292-341, and in the method section (outcomes)we gave the definition for menarche age. 

5. Discussion: line 258: "controlled studies" would not be in the context of an experimental design. Please use a more appropriate term. 

R. We have updated this term for observational studies

6. Conclusion: Considering that most of the evidence is from observational studies, there is a need to revise your statement here ("negatively alter" implies effect which is consistent with your current aim, but not appropriate based on where you got your evidence from, ie. observational studies).

R. We have updated the conclusion, and changed the term effect for association

7. Tables and Figures: Figure 1: --"Trabalhos excluidos" need to be translated to English. R. Ok; --For the box "Full papers evaluated for eligibility), n=16. However, tracked articles (42) minus 29 under Trabalhos excluidos equals 13. R. OK, we have correcte.

 Please correct this. Table 1: --This table would be more informational if the description of the studies were more succinct and if the studyfindings/conclusions were added. For example, "Outcome evaluation age" is redundant since the information is alreadyunder the "Patients" column (Why do you use the term patients instead of "Study Population"?The latter is more appropriate considering the study design of the articles). The exclusion criteria is not helpful--why not add information about variables/confounders that were controlled for by the study?

R. We have updated table 1 according to your suggestions.

 Table 2: --This table is not helpful to my understanding of the quality of evidence. Please make it more stand-alone -- what do the symbols under the column "Certainty of the Evidence" mean? Also, there is no clear explanation in the text about how you came up with the quality of evidence. You only referred the reader to the table when discussing your results. I hope this review will be helpful to the authors.

R. R. We have included information about GRADE in lines 232-242, and we have updated the table 2 in order to be more precise.

---

## [Decision Letter · Decision Letter 1]

14 Jan 2021

PONE-D-20-20990R1

The association between soy-based infant diet and the onset of puberty: A systematic review and meta-analysis

PLOS ONE

Dear Dr. Nunes,

Thank you for submitting your manuscript to PLOS ONE. After careful consideration, we feel that it has merit but does not fully meet PLOS ONE’s publication criteria as it currently stands. Therefore, we invite you to submit a revised version of the manuscript that addresses the points raised during the review process.

ACADEMIC EDITOR: Your manuscript has been reviewed by at least one of the initial reviewers, and they have found some points that need to be addressed before this manuscript is considered for publication. Please go through the the reviewer' comments and consider addressing these points, and prepare a revised version.

We look forward to receiving your revised manuscript.

Kind regards,

Ivan D. Florez; MD, MSc, PhD

Academic Editor

PLOS ONE

Additional Editor Comments (if provided):

Your manuscript has been reviewed by at least one of the initial reviewers, and they have found some points that need to be addressed before this manuscript is considered for publication. Please go through the the reviewer' comments and consider addressing these points, and prepare a revised version.

Reviewers' comments:

Reviewer's Responses to Questions

**Comments to the Author**

1. If the authors have adequately addressed your comments raised in a previous round of review and you feel that this manuscript is now acceptable for publication, you may indicate that here to bypass the “Comments to the Author” section, enter your conflict of interest statement in the “Confidential to Editor” section, and submit your "Accept" recommendation.

Reviewer #1: All comments have been addressed

2. Is the manuscript technically sound, and do the data support the conclusions?

Reviewer #1: Partly

3. Has the statistical analysis been performed appropriately and rigorously? 

Reviewer #1: Yes

4. Have the authors made all data underlying the findings in their manuscript fully available?

Reviewer #1: Yes

5. Is the manuscript presented in an intelligible fashion and written in standard English?

Reviewer #1: No

6. Review Comments to the Author

Reviewer #1: the authors made significant improvement to the manuscript and addressed most of the comments. There are few comments need to be addressed:

1- the hight outcome need to be revisited. I notice that there are 2 studies that have reported height at measurement at beginning of puberty and one as a final adult height. 1- Table 1 should specify if the data is in Cm or SD. 2- present the effect estimate of the reported height at measurement at beginning of puberty and one as a final adult height as separate and as combined. 3- convert the effect estimate to absolute data either SD or Cm rather than leaving it as SMD to improve understanding of the results.

2- all meta-analysis figures did not include label for the direction of the effect estimate.

3- across the manuscript the authors mention that they could or couldn't present meta-analysis figure, but the correct wording is to say that meta analysis could or couldn't be performed. The figure is just a away of the results presentation but meta-analysis is a statistical technique.

4- the definition of high soy intake need to be included in the methods section, then in the results you report what sort of intake was there int he groups, and include the actual intake in table 1.

5- the authors need to describe how the reported findings in the risk of bias section from studies had influenced the risk of bias scores for each one. for example, did studies using self-reported pubertal data were given high or low risk of bias score? this paragraph need to be restructured as specific ROB domains, then overall.

6-the opening paragraph of the discussion need to be focused on the study findings and not to reiterate what was mentioned already in the introduction.

7- the GRADE table, please change to the table view " GRADE profile (V2)) to give details on judgments leading to low quality evidence since its not described in the text.

8- the outcome data in the risk of the outcome with no soy bean, how did you come up with the numbers? are these using the risk in the general population or in the reported comparison group from included studies?

please include foot note to explain this.

for example the mean timing of menarche should not be 0, rather it should be 12,7 years

9- in table 1, please highlight the country of the study in a different column rather than combining it with the population, also age & BMI need to be separated in to columns . All ages need to be standardized either years or months. add a column for total number of children followed in each study.

10- the results are a bit confusing with regard to PP, in table 2 it says 40 out of 1000 are expected to have PP in the unexposed group vs 8 out of 1000 are expected to have PP in the exposed group. although not statistically significant effect, but it is suggesting more PP with lack of exposure, that raises suspicion about the results. Siani 2018, was the only study to report PP while the others had 0 events, the children in his study had cows milk allergy. It is important to understand more information about those children such as: what was the other milk types they took, for how long. was the exposure to soy milk limited to the 1st year of life ?

the discussion section for such observation is important and need to highlight all these details. The current discussion is not focused on this element. please consider if leaving this study out of the analysis would be the most appropriate action because those children are considered different than typically other healthy children in terms of type of nutritional intake because of the allergy.

11- in the limitation section please include that future studies need to report physician assessed puberty stage, and the progression of puberty in those children, in addition to : expected mid parental height, parental puberty.

12- need to consider english editing

7. PLOS authors have the option to publish the peer review history of their article (what does this mean?). If published, this will include your full peer review and any attached files.

Reviewer #1: No

---

## [Author Response · Author response to Decision Letter 1]

28 Feb 2021

Dear reviewer

Again, I want to like to thank you for spending time and efforts to improve our review. Your comments have improved it a lot, and we have learned many methodological aspects by your revision. Again, thank you very much.

1- The height outcome needs to be revisited. I notice that there are 2 studies that have reported height at measurement at beginning of puberty and one as a final adult height. 

R. Thank you for the important detail, we have emphasized the height outcome in the manuscript, line 370 to 380, and in the table 1. 

2- Table 1 should specify if the data is in Cm or SD.

R. We have specified 

3- present the effect estimate of the reported height at measurement at beginning of puberty and one as a final adult height as separate and as combined. 

R. We have specified this in the table 3, actually only Duitama presented height at baseline and as effect estimate, the other two only presented the estimate effect

4- convert the effect estimate to absolute data either SD or Cm rather than leaving it as SMD to improve understanding of the results.

R. As Sinai et al presented data only in SD, we have excluded it from the meta-analysis. We performed meta-analysis only with Duitama e Strom, and in Strom we converted inches in centimeters, and we could perform the meta-analysis in MD (figure 4)

5- all meta-analysis figures did not include label for the direction of the effect estimate.

R. Sorry for this, we have included the direction of the effect estimate in the meta-analyses

6- across the manuscript the authors mention that they could or could not present meta-analysis figure, but the correct wording is to say that meta-analysis could or could not be performed. The figure is just an away of the results presentation, but meta-analysis is a statistical technique.

R. Sorry for this, we have corrected this

7- the definition of high soy intake needs to be included in the methods section, then in the results you report what sort of intake was there int he groups and include the actual intake in table 1.

R. Again, thank you for important observation. Unfortunately, during the protocol of this SR we could not define a high soy intake as eligibility criteria. This is because there is not an exact definition of normal soy intake in children. Additionally, our objective was to include studies who participants intake more soy than most population, as people from Asia. We have included the paragraph below in the manuscript (line 122 to 131).

Given that the mean intake of isoflavones varies from country to country and there is no definition of either a mean or high intake of this phytoestrogen in children, we did not use a cut-off point to separate participants with a high intake from those with a low intake. However, the studies were included if the authors used any tool to classify children as high or low consumers. When the mean daily intake of soy and its’ derivatives was not available, we included patients who were known to have a higher consumption of soy than the general population, such as vegetarians and Asian populations. The assessment of soy consumption was determined through dietary records, food frequency questionnaires, or the participant’s use of soy supplements. 

8- the authors need to describe how the reported findings in the risk of bias section from studies had influenced the risk of bias scores for each one. for example, did studies using self-reported pubertal data were given high or low risk of bias score? this paragraph need to be restructured as specific ROB domains, then overall.

R. Actually we could apply Rob 1 only in Duitama, because the other studies were observational. However, the table 4 (Explanations) we have emphasized that this point was important to rate down the quality of evidence in the risk of bias.

9-the opening paragraph of the discussion need to be focused on the study findings and not to reiterate what was mentioned already in the introduction.

R. We have removed what was mentioned already in the introduction.

10- the GRADE table, please change to the table view " GRADE profile (V2)) to give details on judgments leading to low quality evidence since it’s not described in the text.

R. Thank you very much for this suggestion, we have change for Grade profile V2 and we have given details on judgments leading to low quality evidence

11- the outcome data in the risk of the outcome with no soybean, how did you come up with the numbers? are these using the risk in the general population or in the reported comparison group from included studies?

please include foot note to explain this.

for example the mean timing of menarche should not be 0, rather it should be 12,7 years

R. We have reviewed the table of GRADE with the data from the meta-analyses in order to become it clear than the first one.

12- in table 1, please highlight the country of the study in a different column rather than combining it with the population, also age & BMI need to be separated into columns. All ages need to be standardized either years or months. add a column for total number of children followed in each study.

R. We have highlighted in a different column all these data

13- the results are a bit confusing with regard to PP, in table 2 it says 40 out of 1000 are expected to have PP in the unexposed group vs 8 out of 1000 are expected to have PP in the exposed group. although not statistically significant effect, but it is suggesting more PP with lack of exposure, that raises suspicion about the results. Siani 2018, was the only study to report PP while the others had 0 events, the children in his study had cow’s milk allergy. It is important to understand more information about those children such as: what was the other milk types they took, for how long. was the exposure to soy milk limited to the 1st year of life?

the discussion section for such observation is important and need to highlight all these details. The current discussion is not focused on this element. please consider if leaving this study out of the analysis would be the most appropriate action because those children are considered different than typically other healthy children in terms of type of nutritional intake because of the allergy.

R. We have updated these data in the table of GRADE. I have included all details of Sinai 2018 in the lines 251 to 254. We have included this point in the discussion section (line 447 to 452). In the meta-analysis of height Sinai was excluded due to present data only in SD. We have not excluded it from the review but emphasized this as a limitation (line 447 to 452) 

14- in the limitation section please include that future studies need to report physician assessed puberty stage, and the progression of puberty in those children, in addition to: expected mid parental height, parental puberty.

R. Okay, we have included

15- need to consider English editing

R. Okay, we used Editage for English editing

---

## [Decision Letter · Decision Letter 2]

31 Mar 2021

PONE-D-20-20990R2

Association between a soy-based infant diet and the onset of puberty: A systematic review and meta-analysis

PLOS ONE

Dear Dr. Nunes,

Thank you for submitting your manuscript to PLOS ONE. After careful consideration, we feel that it has merit but does not fully meet PLOS ONE’s publication criteria as it currently stands. Therefore, we invite you to submit a revised version of the manuscript that addresses the points raised during the review process.

We look forward to receiving your revised manuscript.

Kind regards,

Ivan D. Florez, MD, MSc, PhD

Academic Editor

PLOS ONE

Journal Requirements:

Additional Editor Comments (if provided):

Your revised version has improved substantially. However, there is at least one issue to consider before considering this manuscript for publication,

Please check reviewer's comments about table 1.

Reviewers' comments:

Reviewer's Responses to Questions

**Comments to the Author**

1. If the authors have adequately addressed your comments raised in a previous round of review and you feel that this manuscript is now acceptable for publication, you may indicate that here to bypass the “Comments to the Author” section, enter your conflict of interest statement in the “Confidential to Editor” section, and submit your "Accept" recommendation.

Reviewer #1: All comments have been addressed

2. Is the manuscript technically sound, and do the data support the conclusions?

Reviewer #1: Yes

3. Has the statistical analysis been performed appropriately and rigorously? 

Reviewer #1: Yes

4. Have the authors made all data underlying the findings in their manuscript fully available?

Reviewer #1: Yes

5. Is the manuscript presented in an intelligible fashion and written in standard English?

Reviewer #1: Yes

6. Review Comments to the Author

Reviewer #1: thank you for addressing the comments

please verify that all data in table 1 are using the same metrics; for example there is still some data in months while others in years.

7. PLOS authors have the option to publish the peer review history of their article (what does this mean?). If published, this will include your full peer review and any attached files.

Reviewer #1: No

---

## [Author Response · Author response to Decision Letter 2]

20 Apr 2021

April 20, 2021

Dear Reviewer 

We have revised the manuscript according to your comment about table 1. We have reviewed table 1 and table 2, and we used the same metrics (years) in all anthropometric data, as well as follow up time in the table 4. 

Again, thank you so much for spending time reviewing and consequently improving our manuscript.

Your Sincerely,

Vania dos Santos Nunes Nogueira

Department of Internal Medicine – Botucatu Medical School – Sao Paulo State University/UNESP, Sao Paulo, Brazil. Tel: (55 14) 3880 11 71E-mail: vania.nunes-nogueira@unesp.br

---

## [Editor Report · Decision Letter 3]

23 Apr 2021

Association between a soy-based infant diet and the onset of puberty: A systematic review and meta-analysis

PONE-D-20-20990R3

Dear Dr. Nunes,

We’re pleased to inform you that your manuscript has been judged scientifically suitable for publication and will be formally accepted for publication once it meets all outstanding technical requirements.

Kind regards,

Ivan D. Florez, MD, MSc, PhD

Academic Editor

PLOS ONE
---

## [Editor Report · Acceptance letter]

10 May 2021

PONE-D-20-20990R3 

Association between a soy-based infant diet and the onset of puberty: A systematic review and meta-analysis 

Dear Dr. Nunes-Nogueira:

I'm pleased to inform you that your manuscript has been deemed suitable for publication in PLOS ONE. Congratulations! Your manuscript is now with our production department. 

Kind regards, 

on behalf of

Dr. Ivan D. Florez 

Academic Editor

PLOS ONE